# Surgical procedures in Danish children 1999–2018

**Andreas Jensen** [1]*, **Gorm Greisen**[2,3], **Thomas Hjuler** [4], **Lone Graff Stensballe**[1,3]

**1** Department of Paediatrics and Adolescent Medicine, Rigshospitalet, Copenhagen University Hospital, Copenhagen, Denmark, **2** Department of Neonatology, Rigshospitalet, Copenhagen University Hospital, Copenhagen, Denmark, **3** Institute of Clinical Medicine, University of Copenhagen, Copenhagen, Denmark, **4** Department of Otorhinolaryngology and Audiology, Rigshospitalet, Copenhagen University Hospital, Copenhagen, Denmark

* andreas.jensen.01@regionh.dk

## Abstract

### Objective

To assess if the overall utilisation of surgery in Danish children 0–5 years of age increased in the period 1999–2018 in line with the development within specialised medical services. The epidemiology on surgical procedures is scarce.

### Methods

National register-based cohort study of all Danish children born 1994–2018 (n = 1,599,573) using data on surgery in public and private hospitals from The National Patient Register and data on surgery in private specialist practice from The Health Service Register. Incidence rate ratios were calculated using Poisson regression with 1999 as the reference year.

### Results

During the study period 115,573 different children (7.2% of the cohort) underwent surgery. The overall incidence of surgical procedures was stable, but the use of surgery increased in neonates mainly due to an increase in frenectomy. Boys underwent more surgery than girls. In children with severe chronic disease the rate of surgery decreased in public hospitals and increased in private specialist practices.

### Conclusion

The utilisation of surgical procedures in Danish children 0–5 years of age did not increase from 1999 to 2018. The use of available register data in the present study may inspire surgeons to conduct further studies to enhance the knowledge within the area of surgical procedures.

**Data Availability Statement:** Restrictions imposed by Danish law to protect patient privacy imply that the data sets used in the present study are only available through a trusted third party – in this case Statistics Denmark. Statistics Denmark is a state

organisation that holds the data used for the study. It is possible for Danish scientific organisations to be authorised to access data on a Statistics Denmark server which enables individual scientists from these organisations to analyse the data. Similar data requests for data from Statistics Denmark can be applied for through the following link: http://www.dst.dk/en/OmDS/organisation/TelefonbogOrg.aspx?kontor=13&tlfbogsort=sektion.

**Funding:** The authors received no specific funding for this work.

**Competing interests:** The authors have declared that no competing interests exist.

## Introduction

In a previous series of papers we described the use of health service utilisation in Danish children under five years of age [1–3]. The papers highlighted the fact that during the period 1999–2016, the total number of contacts with the health system for children under 5 years of age remained stable, but the services became increasingly specialised. For example, the rate of inpatient hospitalisations increased by 26 percent and the rate of contacts with medical specialists increased by 43 percent [1]. Thus, while the under-five mortality was halved, the increase in specialised health service utilisation increased markedly—also in children without severe chronic disease.

Surgery, however, was not covered in these studies and the utilisation of surgery in children has not been much studied in general. In particular the epidemiology on overall surgery including all specialities is scarce [4, 5]. The procedures of surgery and especially anaesthesia have become more manageable over the years, and the survival of children with severe disease or malformation has improved. Thus, more children may need (repeated) surgery which could lead to increased demand. However, decreasing incidences of different surgical diseases have been found in other countries [6, 7].

Hence, it is of interest to assess whether the overall utilisation of surgery has increased in line with the development within specialised medical services [1]. Thus, the present register-based cohort study in Danish children aged 0–5 years over the 20-year period 1999–2018 was conducted.

## Methods

### Data

The study was based on data from Danish health registers and registers from Statistics Denmark. The individuals were identified using the Danish Civil Registration system [8]. The personal identification numbers were pseudonymised by Statistics Denmark before the authors accessed the data. Data from Statistics Denmark included information on deaths and migrations. The National Patient Register [9] and The Health Service Register [10] were identified as relevant health registers for the purpose of the study. The former contains data on the activity in the public and private hospital whereas the latter contains data on private specialist practice services. The Health Service Register also serves the administrative purpose of registering reimbursements. In Denmark, general health care including the surgical procedures analysed in the present study are provided for free.

### Follow-up

The cohort was based on all Danish children liveborn in the period 1994–2018. Children were followed from birth until five years of age, 31 December 2018, migration, or death, whichever came first. The observation period was chosen as 1999 through 2018 to ensure similar age distributions across calendar years. Before 1999 only children less than 5 years of age were present in the cohort.

### Outcomes

The present study investigated the incidence of surgical procedures in Danish children under five years of age based on register data. Three overall outcomes were considered: 1) surgical procedures in children admitted to public hospitals, 2) surgical procedures in children admitted to private hospitals, and 3) records of general anaesthesia in private specialist practice (outpatient care). Information on surgical procedures performed in public hospitals was obtained from The National Patient Register. All surgical procedures performed in public hospitals in

**Table 1. Categorisation of procedure codes to define surgical specialties.**

| Surgical specialty | NOMESCO codes | NOMESCO code text |
|---|---|---|
| Neurosurgery | KAA-KAW | Nervous system |
| Otorhinolaryngology (ENT) | KBA-KBB; KBD; KDA-KDW; KEHA-KEHB; KGB; KQA; KUD-KUE | Thyroid gland-Parathyroid gland; Carotid body; Ear, nose and larynx; Incision and biopsy of palate; Excision of palate; Trachea; Skin (head and neck); Transluminal endoscopy of ear, nose and larynx; Transluminal endoscopy of mouth and pharynx |
| Paediatric surgery | KBC; KBW; KJA-KJW, KKA-KKW; KUJ-KUK | Adrenal gland; Reoperations in endocrine surgery; Digestive system and spleen; Urinary system, male genital organs and retroperitoneal space; Transluminal endoscopy of gastrointestinal tract-Transluminal endoscopy of urinary tract-Transluminal endoscopy of female genital tract |
| Ophthalmology (Eye) | KCA-KCW; | Eye and adjacent structures |
| Plastic surgery | KEA; KEHC; KQB-KQX | Lips; Reconstructive operations on palate; Skin (trunk, upper limb, lower limb, unspecified region, reoperations on skin) |
| Oral/maxillofacial surgery | KEB-KEG; KEHK-KEHW | Teeth; Gingiva and alveoli; Mandible-Maxilla; Miscellaneous operations on jaws; Mandibular joint; Removal of implant or external fixation device from palate |
| Cardiothoracic surgery | KFA-KFX; KGA; KGC-KGW; KUG | Heart and major thoracic vessels; Chest wall, pleura and diaphragm; Bronchus; Lung; Mediastinum; Reoperations in thoracic surgery; Transluminal endoscopy of trachea, bronchus and pleura |
| Gynaecology | KLA-KLW; KUL | Female genital organs; Transluminal endoscopy of female genital tract |
| Orthopaedic surgery | KNA-KNW | Musculoskeletal system |
| Vascular surgery | KPA-KPX | Peripheral vessels and lymphatic system |
| Minor surgical procedures | KT | Minor surgical procedures |

Denmark are recorded in the register according to the NOMESCO classification. The NOMESCO classification is based on the traditions of the surgical profession in the Nordic countries and reflects surgical practice in these countries, arranging surgical procedures according to functional-anatomic body system [11]. As the NOMESCO classification is only used in the Nordic countries we categorised the surgical procedures according to surgical specialty to address an international audience (Table 1). Information on local or general anaesthesia regarding the procedures in public hospitals was not available. Specific descriptions of how records in both public and private hospitals were identified can be found in the supporting information (S1 Text).

The specialties of obstetrics (KMA-KMW) and breast surgery (KH) were also considered, but the number of observations was deemed too low for further analysis with only 167 and 46 surgical interventions in hospitalised patients under five years of age during the entire study period.

However, note that information on private hospitals was only included in The National Patient Register from 2002 onwards and due to initial lack of registrations in the subsequent years, the present study covered the period 2004–2018 for the outcome surgical procedures in private hospitals.

Surgical interventions in medical specialist practice were defined using records of general anaesthesia in the Health Service Register with a 2-digit specialty code value of 01 corresponding to a service of anaesthesiology. All residents in Denmark have access to the public healthcare system, and most services are provided free of charge. In the primary healthcare system, a number of private practicing specialists provide surgery in general anaesthesia, however in child surgery almost exclusively among otologists.

## Statistics

The results of the present study are illustrated by the incidence of surgical procedures plotted against calendar year. The incidence was defined as the number of surgical procedures per

thousand person-years. The numbers on which the figures are based can be found in (Tables A-E in S1 File). The results are presented by calendar year as totals and by sex (female, male), age group (< 28 days, 28–365 days, 1–5 years) and chronic disease status (yes, no). The definition of chronic disease is described in prior studies [2, 12, 13]. In the present study it was included as a time-varying variable with the status updated at the beginning of each calendar year. The incidence rates and incidence rate ratios were estimated using Poisson regression. The ratios were obtained from Poisson regression models with 1999 as the reference year and were denoted as $IRR_{2000}$, $IRR_{2001}$, ..., $IRR_{2018}$. The logarithm of the number of person-years was included as a model term with the coefficient constrained to 1. The crude calendar year-specific rates were obtained by fitting the corresponding model without an intercept term. Each child was allowed to contribute with multiple events during follow-up, but not with more than one event on the same day within a given analysis. The 95% confidence limits for the figures were obtained by using the jackknife resampling technique with recurrent events among individuals as clusters.

The number of total surgical procedures and the numbers within each surgical specialty are presented in the results section. To provide information on the composition of the different surgical specialties the five most frequently occurring procedure codes within each surgical specialty are presented in the supporting information (Table F in S1 File).

Data management and analyses were performed using Stata 17.0.

## Results

The size of the cohort amounted to 1,599,573 children of which 115,213 (7.2%) underwent at least one surgical procedure in a public hospital during the observation period 1999–2018. The overall number of surgical procedures in public hospitals was 177,015 among children aged 0–5 years. The surgical procedures were distributed as 87,098 (75.6% of the cases) children with exactly 1 procedure, 24,210 (21.0%) children with 2–4 procedures, and 3,905 (3.4%) children with at least 5 procedures.

In private specialised practise 339,200 different children (21.2% of the cohort) experienced 582,046 events of general anaesthesia. Only 809 different children experienced a total of 839 surgical procedures in private hospitals (2002–2018).

The total number of surgical procedures in public hospitals by surgical specialties are presented in Table 2.

**Table 2. Number of surgical procedures in public hospitals by surgical specialty, Danish children 0–5 years of age, 1999–2018.**

| Surgical specialty | Number of children with at least one event | Number of events |
|---|---|---|
| Otorhinolaryngology (ENT) | 48,849 | 62,191 |
| Paediatric surgery | 31,201 | 43,240 |
| Minor surgical procedures | 14,972 | 22,944 |
| Orthopaedic surgery | 15,807 | 21,520 |
| Cardiothoracic surgery | 8275 | 11,979 |
| Plastic surgery | 8089 | 11,648 |
| Neurosurgery | 2377 | 4758 |
| Ophthalmology (eye) | 3581 | 4,694 |
| Vascular surgery | 967 | 1071 |
| Oral and maxillofacial surgery | 879 | 943 |
| Gynaecology | 539 | 605 |
| **Total** | 115,213 | 177,015 |

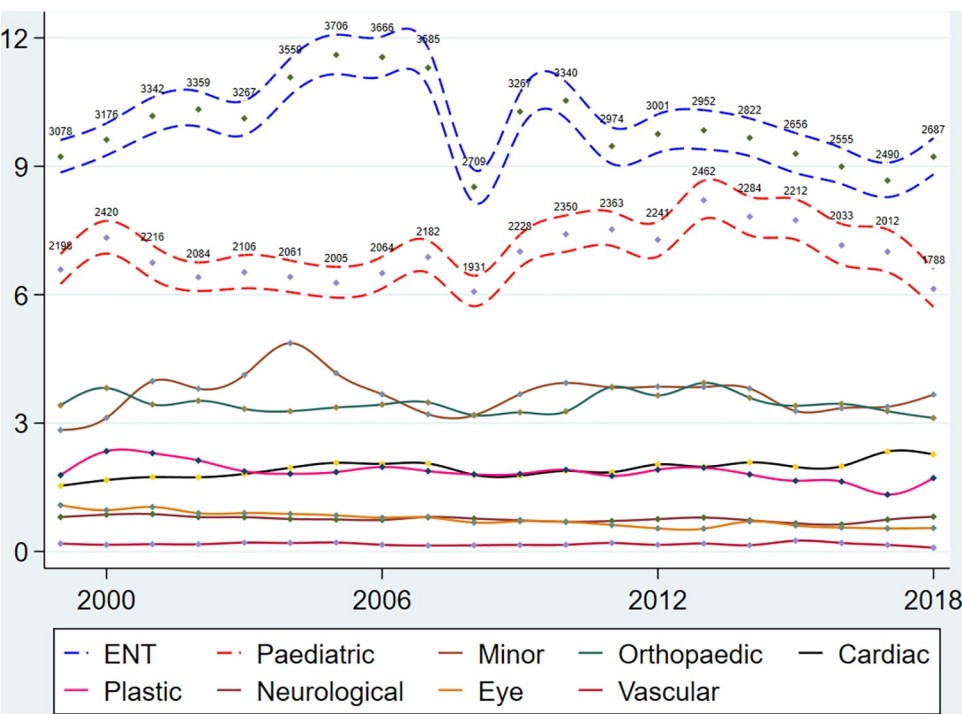

**Fig 1. Surgical procedures in public hospitals per 1000 person-years, Danish children 0–5 years of age, 1999–2018.**
The curves represent the surgical specialties (sorted by incidence in the legend). The 95% pointwise confidence limits and the absolute numbers are only provided for the most frequent procedures to enhance the visual appearance.

The five most frequent surgical procedure codes within each surgical specialty are presented in the supporting information (Table F in S1 File).

The incidence of surgical procedures in public hospitals was stable over time (Figs 1 and 2). Note that the two surgical specialties of oral and maxillofacial surgery and gynaecology were omitted from Fig 1 to enhance the visual appearance, but the incidence curves of all specialties, including oral and maxillofacial surgery and gynaecology, are presented in the supporting information.

The largest overall rate was observed in 2004 with 29.61 surgical procedures per 1000 person-years, which corresponded to an increase of 14 percent, $IRR_{2004} = 1.14$ (95% CI: 1.11–1.17), compared to the rate in 1999 (Table A in S1 File). The exception to the stable pattern was a large decrease in surgical activity in 2008. The incidence of surgical procedures decreased from 2012 onwards which ultimately resulted in an overall decrease, since $IRR_{2018} = 0.93$ (95% CI: 0.87–0.99) compared to the rate in 1999 (Table B in S1 File).

From the left panel of Fig 2 it is seen that boys underwent more surgery compared to girls. Furthermore, it is also apparent that the pattern of decreasing surgery in recent years, observed in Fig 1, was especially driven by a decrease among boys, $IRR_{2018} = 0.95$ (95% CI: 0.92–0.99). The overall rate of surgical procedures increased among girls, $IRR_{2018} = 1.09$ (95% CI: 1.04–1.14) cf. Table A in S1 File.

In relative numbers the neonatal age group underwent a substantial increase of 184 percent throughout the study period, $IRR_{2018} = 2.84$ (95% CI: 2.54–3.17). In absolute numbers this corresponded to a level of 430 surgical procedures in neonates in 1999 compared to 1140 procedures in 2018 (Table A in S1 File). As an exploratory finding it was observed that the increase among neonates was partially driven by the increased use of the otorhinolaryngological procedure of tongue frenectomy (KEJC20), cf. Figs B and C in S2 File.

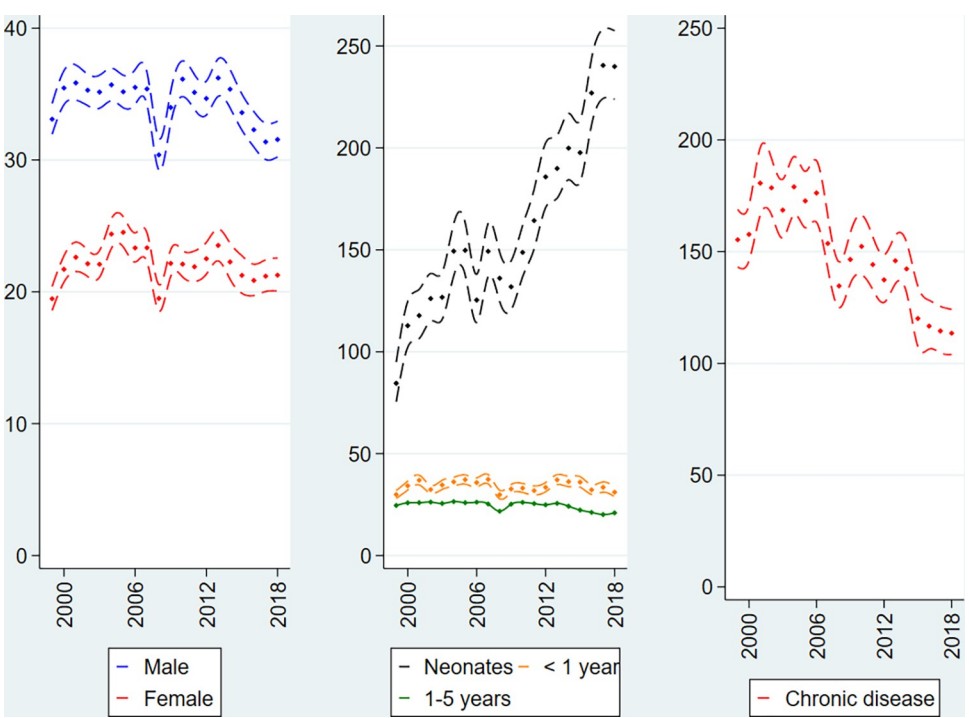

**Fig 2. Overall rates of surgical procedures in public hospitals per 1000 person-years, Danish children 0–5 years of age, 1999–2018.** Sex-specific incidence (left panel), age-specific incidence (middle panel) and chronic disease-specific incidence (right panel) with 95% pointwise confidence limits.

From the right panel of Fig 2 it is seen that the rate of surgical procedures in children with chronic disease decreased during the study period, $IRR_{2018}$ = 0.73 (95% CI: 0.68–0.78). Similar patterns were observed for all surgical specialties (Figs D-N in S2 File).

In the vast majority of cases, 96.1%, general anaesthesia was related to otology. In private specialist practise some changes in incidence rates of surgical procedures, defined by the utilisation of general anaesthesia, were observed during the study period (Fig 3). The rates of surgical procedures increased until around 2010–2012, and after that point in time the incidence decreased. The difference between the sexes was similar to the one observed within public hospitals whereas the age group difference was inverted: the highest utilisation of surgery in private specialist practice was found in the oldest age group (1–5 years of age). Finally, in children with chronic disease, the rate of surgery in private specialist practice increased, $IRR_{2018}$ = 1.28 (95% CI: 1.19–1.38), which was in contrast to the pattern within public hospitals.

## Discussion

In the present national register-based cohort study the use of surgical procedures in more than 1.5 million Danish children 0–5 years of age from 1999 to 2018 was studied. We hypothesised that the use of surgical procedures was increasing. The hypothesis was not supported: there was no clear overall trend of an increased number of children undergoing surgery in Denmark during the last 20 years (Figs 1–3). In fact, a decreasing tendency from 2012 onwards was found in public hospitals as well as in private specialist practice (Figs 2 and 3). The decrease was driven by the development of surgery in boys. In children with chronic disease the development was ambiguous: the use of surgery decreased in public hospitals but increased in private specialist practice. Future studies are warranted to explore these developments.

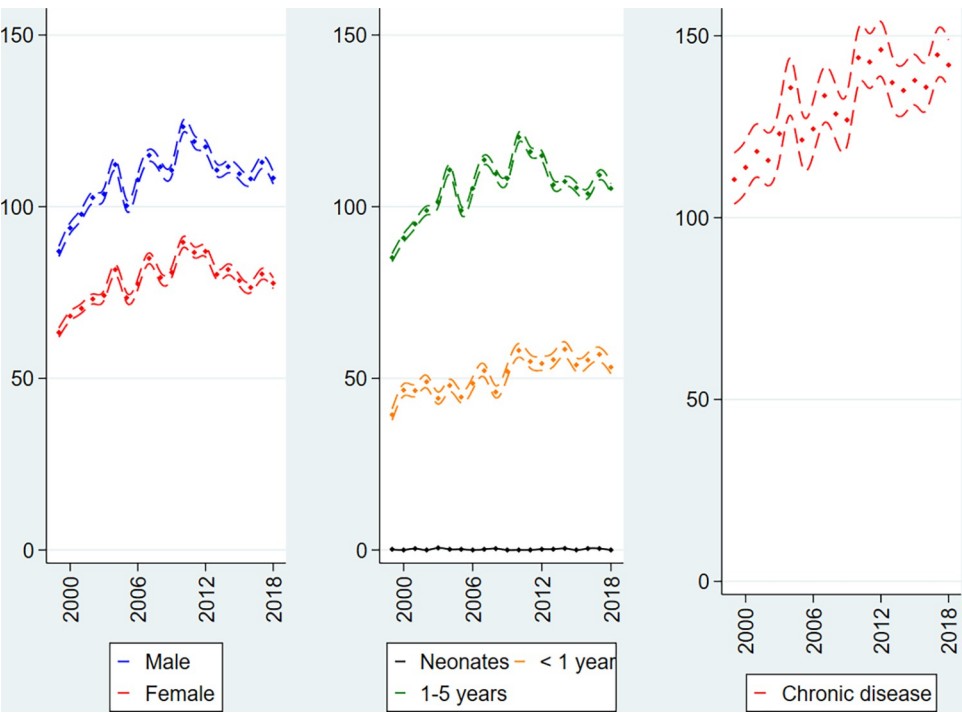

**Fig 3. General anaesthesia in private specialist practice per 1000 person-years in Danish children 0–5 years of age, 1999–2018.** Sex-specific incidence (left panel), age-specific incidence (middle panel) and chronic disease-specific incidence (right panel) with 95% pointwise confidence limits.

Surgery in young children persist to be a substantial decision for parents and surgeons which could partly explain the stable tendency despite the circumstances suggesting an increased demand. The interventions of surgery and anaesthesia are not without risks and post-operative pain management is difficult in young children due to communication barriers. Also, diagnostic methods have improved which may have helped reduce unnecessary surgery.

Public health care in Denmark is under pressure by an increasing mismatch between what can be done within the capacity of available beds, nurses, and doctors and what is requested by the population and politicians. Although children in general are highly prioritised in hospitals, also for surgery, it is possible that the recent years have been characterised by an increased postponement and eventual cancellation. This could also explain the decrease in boys, since the surgical procedures subject to cancellation were more frequent in boys compared to girls. Examples for the reasons why surgical procedures were declining include hypertrophic pyloric stenosis and inguinal hernia [6, 7], and these diseases are overrepresented in boys.

Alternatively, a general increase in child health could also explain a decreased need of surgery in the age group 0–5 years. These explanations are, however, mere speculations that may encourage more targeted studies with the purpose of optimising the use of surgery in young children. In general, it is a worthwhile discussion as to which extent the utilisation of surgery is driven by the demand of the patients or the capacity available within the political and financial circumstances.

Some sharp decreases and increases lasting just one year were observed. To our best knowledge the decline in public hospital surgery in 2008 could be explained by a nurses' strike which affected all areas of the health care system [14–16]. In fact, an even larger and more sustained post-strike catch-up effect (11.5% in 2009) could have been expected. However, this could as

well be attributed to the overall question of capacity in the healthcare system. The cause of the sharp increase in private specialist practice in 2004 is unclear, but this could be explained by changes in registration guidelines leading to the same specific procedure being registered more than once during a short implementation period.

The organisation of health care differs between countries. In Denmark hospital care is mainly provided by public hospitals with few private hospital providers ($< 1\%$ of hospital beds). Accordingly, we observed surgery in young children in Danish private hospitals to be rare. On the other hand, private specialist practice was a major contributor, almost exclusively driven by otology procedures.

The strengths of the present study include the national cohort design and the sample size of around 1.5 million children which enhance generalisability and diminish statistical uncertainty. In addition, a clear hypothesis was readily refuted. Also, some important weakness must be acknowledged. First, we used general anaesthesia as a marker for significant surgery in the private specialist practice. This was done to obtain one uniform measure for the diversity of surgical procedures carried out in private specialist practice. Thus, to study the trend over time, we aimed to identify one uniform measure of surgical activity instead of studying the activity within all surgical specialities in private practice. Second, the data were based on procedure codes included in routine medical care, and the validity and completeness of coding is unknown. In some contexts, codes are used for reimbursement, for instance in private specialist practice, and thus completeness is assumed to be high. On the other hand, changes in reimbursement schemes may cause changes in the coding system if several codes become relevant. Third, physiotherapy and other non-surgical services were coded as procedures which led to our introduction of the additional category of minor surgical procedures.

## Conclusion

In conclusion, the use of surgery did not increase during the period 1999 to 2018 and perhaps the trend was even decreasing from 2012 onwards. The use of available register data in the present study may inspire surgeons to conduct further studies to enhance the knowledge within the area of surgical procedures.

## Supporting information

**S1 File. Tables.**
(DOCX)

**S2 File. Supporting figures.**
(DOCX)

**S1 Text. Description of identification of records.**
(DOCX)

## Author Contributions

**Conceptualization:** Andreas Jensen, Gorm Greisen, Thomas Hjuler, Lone Graff Stensballe.

**Formal analysis:** Andreas Jensen.

**Methodology:** Andreas Jensen, Gorm Greisen, Thomas Hjuler.

**Project administration:** Lone Graff Stensballe.

**Software:** Andreas Jensen.

**Supervision:** Gorm Greisen, Thomas Hjuler, Lone Graff Stensballe.

**Validation:** Thomas Hjuler.

**Writing – original draft:** Andreas Jensen.

**Writing – review & editing:** Gorm Greisen, Thomas Hjuler, Lone Graff Stensballe.

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
