## [Decision Letter · Decision Letter 0]

6 Mar 2023

PONE-D-22-35565False alarm: surgical procedures in Danish children has not increased.PLOS ONE

Dear Dr. Jensen,

Thank you for submitting your manuscript to PLOS ONE. After careful consideration, we feel that it has merit but does not fully meet PLOS ONE’s publication criteria as it currently stands. Therefore, we invite you to submit a revised version of the manuscript that addresses the points raised during the review process.Dear authors, please strictly follow comments and remarks from the first reviewer who has very concrete comments and remarks to be amended.Please strictly follow the comments of the  first reviewer who proposed major revision.Please submit your revised manuscript by Apr 20 2023 11:59PM. If you will need more time than this to complete your revisions, please reply to this message or contact the journal office at plosone@plos.org. Please include the following items when submitting your revised manuscript:A rebuttal letter that responds to each point raised by the academic editor and reviewer(s). You should upload this letter as a separate file labeled 'Response to Reviewers'.A marked-up copy of your manuscript that highlights changes made to the original version. You should upload this as a separate file labeled 'Revised Manuscript with Track Changes'.An unmarked version of your revised paper without tracked changes. You should upload this as a separate file labeled 'Manuscript'.

We look forward to receiving your revised manuscript.

Kind regards,

Stefan Grosek, Ph.D., M.D.,

Academic Editor

PLOS ONE

Journal Requirements:

Additional Editor Comments:

Dear authors

Your study was reviewed by two reviewers which found it as very interesting, however one of the reviewer have many comments and remarks which should be considered while preparing a revised manuscript.

Reviewers' comments:

Reviewer's Responses to Questions

**Comments to the Author**

1. Is the manuscript technically sound, and do the data support the conclusions?

Reviewer #1: Partly

Reviewer #2: Yes

2. Has the statistical analysis been performed appropriately and rigorously? 

Reviewer #1: I Don't Know

Reviewer #2: Yes

3. Have the authors made all data underlying the findings in their manuscript fully available?

Reviewer #1: Yes

Reviewer #2: Yes

4. Is the manuscript presented in an intelligible fashion and written in standard English?

Reviewer #1: Yes

Reviewer #2: Yes

5. Review Comments to the Author

Reviewer #1: Jensen and colleagues present a study using the well-described Danish administrative data to investigate the number of surgical procedures conducted in Danish children below the age of six years between 1999 and 2018. They used the same statistical approach as described in their preceding studies (references 1-3). They investigated both data from the National Patient Register for inpatients and the Health Services Register in order to investigate the number of surgical procedures conducted in outpatients using the surrogate code of general anaesthesia. They concluded that the number of surgical procedures did not increase.

The manuscript is generally well-written, although the non-agreement of the subject and the auxilliary verb in the title particularly incommodes me, but would benefit from some attention to detail with respect to language and the use of commas to ease reading. However,

I have several aspects for the authors to consider for the revision.

Major:

1. The most important issue for me are the definitions of surgical procedures. I did review the NOMESCO-classification of surgical procedures as I was irritated by the distribution of procedures in Table F of the Supplemental information 1. While reading the study, I wondered why would there be so many vascular surgery procedures. After I had a look at the NOMESCO-classification, I did know why: Procedures at the lymphatic system were classified conjointly with those in arteries and veins. On page 14, the referenced document of the NOMESCO-classification states that "The way in which a procedure is related to the functional-anatomical axis determines the position of the code within the classification" and one the very same page, it states that "The procedure codes are independent of surgical speciality". The present study did use the NOMESCO-classification to assign the procedure codes to a speciality (Table 1 on line 81) for which the classification is not intended. I may exemplify my point with some examples, besides those already mentioned for lymph node exstirpation, which is more of a paediatric or general surgery procedure. The most frequent cardiothoracic surgery operation is the insertion of a chest tube, but only two cardiothoracic surgical departments participate in the treatment of congenital heart diseases (according to DOI: 10.1016/j.jacc.2017.03.587). So if a neonate would require a chest tube, these departments send a flying cardiothoracic surgeon to a neontal department? I suppose that this would be rather ineffective for a tension pneumothorax, so who performs these procedures? In my country, it would either be the neonatologist or the paediatric surgeon. On the other hand, the most frequent paediatric surgical operation was inguinal hernia: However, in 2014 Bisgaard et al. (DOI: 10.1007/s10029-013-1077-8) described that dedicated paediatric surgical units typically operate on children below 24 months in Denmark due to the centralisation of anaesthetic care in this age group, whereas inguinal hernia repair above this age limit is typically performed by general surgeons (as it is surgery in children, but not necessarily paediatric surgery). Another example from Plastic Surgery would be the correction of cleft palate malformations, which are in the author's institution indeed performed by a specialised unit within the Plastic Surgery department, but would be performed by maxillofacial surgeons in Aarhus (https://www.en.auh.dk/departments/department-of-oral-and-maxillofacial-surgery/gl-forside/research/cleft-lip-and-palate/). Another issue is the category of minor procedures: While lumbar tap and bladder puncture are procedures conducted by paediatricians, which even makes it difficult to summarise them as surgery at all, there are major procedures among this category: For example cholangiograms (KTJK) or cardiac catheterisations, which represent more than 5% of all minor procedures (Table F in S1). Of note, these are only examples of the inconsistencies that is introduced by applying the anatomical classification in order to define specialty care. Without clear adjustments according to a competence or training catalogue of the individual specialties, any points that reference surgical procedures with respect to specialty cannot be made. This has to be corrected if any conclusion were to be made with respect to surgical specialties.

2. I doubt the suitability of the surrogate parameter of general anaesthesia as a marker of a surgical procedure. At first, it is unclear which independent private practices exist and with which specialty. As these information are not provided within the manuscript, the reader might be tempted to assume that the situation would be similar to the readers own country. This would result in a very distorted interpretation: While the number of private practices in England might be limited, in Germany you would find a private practice for almost every specialty (except cardiac surgery). Without any information on the situation in Denmark, the data is uninterpretable. Moreover, the state's system centralises anaesthetic care for children below 24 months, but even a neonate might be operated in a private practice? Moreover, one is likely to assume that many of these procedures might be dental care, which I would not consider surgery. Consequently, if the authors just state that they had knowledge on the type of surgery in the discussion (lines 208-210), this is neither appropriate nor have the drawbacks been adequately discussed.

3. The authors used the same statistical method as in their preceding papers (reference 1-3). However, I have some doubts if this is adequate. In their preceding works, they investigated aspects that might be assumed to be independent, but with surgical procedures one expects some effect of autocorrelation due to secondary procedures that are inevitably linked to primary ones (e.g. stoma closure following enterostomy, two-step repairs in congenital heart diseases etc). Has this been investigated? Likewise, was overdispersion of the model assessed? The numbers reported on children with multiple procedures (lines 119-121) indicate that these aspects might be problematic. I did note that on the preceding reports, a Biostatistician was among the authors, but not in the present work. Please comment.

4. It remains unclear why the study period was chosen and why procedures in private hospitals were included only with their availability in the database. Wouldn't it be the more natural choice to start the investigation in 2002 then, because lines 68-70 do not provide a strong justification for the choice of the investigated years.

5. The introduction and the discussion are way to unidimensional: Reference 5 states that the rate of surgeries was stable and there is a rich literature on the decrease of incidences of surgical diseases in all age groups, for example in hypertrophic pyloric stenosis (DOI: 10.1111/ans.15377) or inguinal hernia (DOI: 10.4174/astr.2019.97.1.41). Consequently, the investigations that the authors suggest that their work might initiate (lines 228-229) are already done. Meanwhile, the elephant in the room, surgical care in infants and toddlers being more focused on complex conditions with repeated procedures, is not addressed.

6. The discussion in lines 188-198 is not adequate. At first, the pandemic has not been included in the investigations and implying similarities on the base of speculation is not helpful. Examples for the reasons why surgical procedures were declining are provided in my comment #5 and these diseases do overproportionately affect boys (which is why procedures in boys were more often cancelled, because they are more often operated on due to a male predominance in many surgical diseases).

7. Moreover, it is not discussed at all why the nurse strike in 2008 did not result in a catch-up effect of postponed elective surgeries in the years afterwards, although the reader would expect such a point to be discussed.

Minor:

8. It is unclear to me why the overall numer of procedure is included in Figure 1, especially with total numbers, if the graph focuses on the discription of procedures by "specialty" per 1,000 person years and the data may be obtained from the supplemental data anyway.

9. The overall incidence by combining the separate incidences of males and females in Panels 2A/3A (and the subsequent panels of the Figures in S2) does not add any information, because these populations are not similar based on the sex-specific incidence and should be ommitted.

10. The figures are not designed to be inclusively. The use of colours is certainly not colourblind-friendly and needs to be adjusted or at least supplemented by different line types.

11. Connecting individual values with a line results in the reader to assume that they are connected in any way. However, the statistical analysis assumes them to be independent, which would favour a depiction of individual data points without being connected by a line. The use of individual confidence intervals (instead of a, for example shaded region) also supports not using a connecting line.

12. The colour choice of the confidence intervals together with the rather low image quality makes the visual assessment of the confidence intervals quite difficult.

13. Why does Figure 1 not have any confidence intervals apart from those of the overall data?

Reviewer #2: Very well written paper with detailed analysis.

In line 180 and 193 - you have mentioned "decrease in children" - can you please substantiate and provide references?

Frenulotomy - Can you please elaborate if these are performed in GA?

6. PLOS authors have the option to publish the peer review history of their article (what does this mean?). If published, this will include your full peer review and any attached files.

Reviewer #1: **Yes: **Christina Oetzmann von Sochaczewski

Reviewer #2: **Yes: **Mr. Ashish Desai

---

## [Author Response · Author response to Decision Letter 0]

3 Apr 2023

Review Comments to the Author and Author response

Reviewer #1: Jensen and colleagues present a study using the well-described Danish administrative data to investigate the number of surgical procedures conducted in Danish children below the age of six years between 1999 and 2018. They used the same statistical approach as described in their preceding studies (references 1-3). They investigated both data from the National Patient Register for inpatients and the Health Services Register in order to investigate the number of surgical procedures conducted in outpatients using the surrogate code of general anaesthesia. They concluded that the number of surgical procedures did not increase.

The manuscript is generally well-written, although the non-agreement of the subject and the auxilliary verb in the title particularly incommodes me, but would benefit from some attention to detail with respect to language and the use of commas to ease reading.

Author response: Thank you for the thorough review of our manuscript. We have changed the title to ‘Surgical procedures in Danish children 1999-2018’.

However, I have several aspects for the authors to consider for the revision.

Major:

1. The most important issue for me are the definitions of surgical procedures. I did review the NOMESCO-classification of surgical procedures as I was irritated by the distribution of procedures in Table F of the Supplemental information 1. While reading the study, I wondered why would there be so many vascular surgery procedures. After I had a look at the NOMESCO-classification, I did know why: Procedures at the lymphatic system were classified conjointly with those in arteries and veins. On page 14, the referenced document of the NOMESCO-classification states that "The way in which a procedure is related to the functional-anatomical axis determines the position of the code within the classification" and one the very same page, it states that "The procedure codes are independent of surgical speciality". The present study did use the NOMESCO-classification to assign the procedure codes to a speciality (Table 1 on line 81) for which the classification is not intended. I may exemplify my point with some examples, besides those already mentioned for lymph node exstirpation, which is more of a paediatric or general surgery procedure. The most frequent cardiothoracic surgery operation is the insertion of a chest tube, but only two cardiothoracic surgical departments participate in the treatment of congenital heart diseases (according to DOI: 10.1016/j.jacc.2017.03.587). So if a neonate would require a chest tube, these departments send a flying cardiothoracic surgeon to a neontal department? I suppose that this would be rather ineffective for a tension pneumothorax, so who performs these procedures? In my country, it would either be the neonatologist or the paediatric surgeon. On the other hand, the most frequent paediatric surgical operation was inguinal hernia: However, in 2014 Bisgaard et al. (DOI: 10.1007/s10029-013-1077-8) described that dedicated paediatric surgical units typically operate on children below 24 months in Denmark due to the centralisation of anaesthetic care in this age group, whereas inguinal hernia repair above this age limit is typically performed by general surgeons (as it is surgery in children, but not necessarily paediatric surgery). Another example from Plastic Surgery would be the correction of cleft palate malformations, which are in the author's institution indeed performed by a specialised unit within the Plastic Surgery department, but would be performed by maxillofacial surgeons in Aarhus (https://www.en.auh.dk/departments/department-of-oral-and-maxillofacial-surgery/gl-forside/research/cleft-lip-and-palate/). Another issue is the category of minor procedures: While lumbar tap and bladder puncture are procedures conducted by paediatricians, which even makes it difficult to summarise them as surgery at all, there are major procedures among this category: For example cholangiograms (KTJK) or cardiac catheterisations, which represent more than 5% of all minor procedures (Table F in S1). Of note, these are only examples of the inconsistencies that is introduced by applying the anatomical classification in order to define specialty care. Without clear adjustments according to a competence or training catalogue of the individual specialties, any points that reference surgical procedures with respect to specialty cannot be made. This has to be corrected if any conclusion were to be made with respect to surgical specialties.

Author response: We acknowledge the reviewer’s insightful comments about the classification of procedures. The NOMESCO classification is based on the traditions of the surgical profession in the Nordic countries and reflects surgical practice in these countries, arranging surgical procedures according to functional-anatomic body system. As the NOMESCO classification is only used in the Nordic countries we categorised the surgical procedures according to surgical specialty to address an international audience. This has been added to the revised manuscript (lines 82-86). Further, we provided an additional column in Table 1 containing short explanations of each category.

Manuscript change: The NOMESCO classification is based on the traditions of the surgical profession in the Nordic countries and reflects surgical practice in these countries, arranging surgical procedures according to functional-anatomic body system [11]. As the NOMESCO classification is only used in the Nordic countries we categorised the surgical procedures according to surgical specialty to address an international audience (Table 1)

2. I doubt the suitability of the surrogate parameter of general anaesthesia as a marker of a surgical procedure. At first, it is unclear which independent private practices exist and with which specialty. As these information are not provided within the manuscript, the reader might be tempted to assume that the situation would be similar to the readers own country. This would result in a very distorted interpretation: While the number of private practices in England might be limited, in Germany you would find a private practice for almost every specialty (except cardiac surgery). Without any information on the situation in Denmark, the data is uninterpretable. Moreover, the state's system centralises anaesthetic care for children below 24 months, but even a neonate might be operated in a private practice? Moreover, one is likely to assume that many of these procedures might be dental care, which I would not consider surgery. Consequently, if the authors just state that they had knowledge on the type of surgery in the discussion (lines 208-210), this is neither appropriate nor have the drawbacks been adequately discussed.

Author response: In the revised manuscript (lines 99-101 and line 173) we have provided information about the medical specialties related to general anaesthesia. All residents in Denmark have access to the public healthcare system, and most services are provided free of charge. In the primary healthcare system, a number of private practicing specialists provide surgery in general anaesthesia, however in child surgery almost exclusively among otologists. Specifically, we found 96.1% of the cases of general anaesthesia to be related to otology. None of the records were related to dental care.

Manuscript changes: In the vast majority of cases, 96.1%, general anaesthesia was related to otology (…) All residents in Denmark have access to the public healthcare system, and most services are provided free of charge. In the primary healthcare system, a number of private practicing specialists provide surgery in general anaesthesia, however in child surgery almost exclusively among otologists.

3. The authors used the same statistical method as in their preceding papers (reference 1-3). However, I have some doubts if this is adequate. In their preceding works, they investigated aspects that might be assumed to be independent, but with surgical procedures one expects some effect of autocorrelation due to secondary procedures that are inevitably linked to primary ones (e.g. stoma closure following enterostomy, two-step repairs in congenital heart diseases etc). Has this been investigated? Likewise, was overdispersion of the model assessed? The numbers reported on children with multiple procedures (lines 119-121) indicate that these aspects might be problematic. I did note that on the preceding reports, a Biostatistician was among the authors, but not in the present work. Please comment.

Author response: It is true that the observations cannot be assumed to be independent. Thus, we have specified that the dependence between recurrent events within the same individuals has been accommodated by using the jackknife resampling technique for the calculation of the confidence limits (lines 115-116).

Poisson regression is merely used as a tool to calculate the incidence rates and rate ratios by including the logarithm of the number of person-years as a model term with the coefficient constrained to 1 (lines 111-112). Thus, the model does not assume the outcome variable to follow a Poisson distribution. In fact, this model is just a special case of the proportional hazard models where the baseline hazard function is piecewise constant.

We agree that children with multiple procedures is an important aspect. However, it is also worth noting that 75% of the case children experience only on surgical procedure, and only approximately 3% experience more than four procedures.

The first author holds a PhD degree in biostatistics from University of Copenhagen.

Manuscript changes: The 95% confidence limits for the figures were obtained by using the jackknife resampling technique with recurrent events among individuals as clusters (…) The logarithm of the number of person-years was included as a model term with the coefficient constrained to 1.

4. It remains unclear why the study period was chosen and why procedures in private hospitals were included only with their availability in the database. Wouldn't it be the more natural choice to start the investigation in 2002 then, because lines 68-70 do not provide a strong justification for the choice of the investigated years.

Author response: The study period was chosen as 1999-2018 due to data availability and to ensure that all calendar years throughout the study period included children 0-5 years of age. Cf. line 70-74. Apart from that, however, we used the maximal study period to exploit as much information as possible.

We acknowledge that the registration of procedures in private hospitals is unclear. We have tried to specify in the revised version (lines 94-96).

Manuscript changes: The cohort was based on all Danish children liveborn in the period 1994-2018. Children were followed from birth until five years of age, 31 December 2018, migration, or death, whichever came first. The observation period was chosen as 1999 through 2018 to ensure similar age distributions across calendar years (…) However, note that information on private hospitals was only included in The National Patient Register from 2002 onwards and due to initial lack of registrations in the subsequent years, the present study covered the period 2004-2018 for the outcome surgical procedures in private hospitals.

5. The introduction and the discussion are way to unidimensional: Reference 5 states that the rate of surgeries was stable and there is a rich literature on the decrease of incidences of surgical diseases in all age groups, for example in hypertrophic pyloric stenosis (DOI: 10.1111/ans.15377) or inguinal hernia (DOI: 10.4174/astr.2019.97.1.41). Consequently, the investigations that the authors suggest that their work might initiate (lines 228-229) are already done. Meanwhile, the elephant in the room, surgical care in infants and toddlers being more focused on complex conditions with repeated procedures, is not addressed.

Author response: We very much appreciate the valuable comments from the reviewer regarding complex conditions. In fact, we have adopted another aspect by defining children with chronic disease and presented the rates in that group separately. We also included the reviewer’s input in the introduction (lines 51-52). 

Manuscript change: However, decreasing incidences of different surgical diseases have been found in other countries.

6. The discussion in lines 188-198 is not adequate. At first, the pandemic has not been included in the investigations and implying similarities on the base of speculation is not helpful. Examples for the reasons why surgical procedures were declining are provided in my comment #5 and these diseases do overproportionately affect boys (which is why procedures in boys were more often cancelled, because they are more often operated on due to a male predominance in many surgical diseases).

Author response: Again, we would like to thank the reviewer for this useful comment. We agree and the discussion has been revised accordingly (lines 208-210). Also, the mention of the pandemic has been deleted.

Manuscript change: Examples for the reasons why surgical procedures were declining include hypertrophic pyloric stenosis and inguinal hernia [6,7], and these diseases are overrepresented in boys.

7. Moreover, it is not discussed at all why the nurse strike in 2008 did not result in a catch-up effect of postponed elective surgeries in the years afterwards, although the reader would expect such a point to be discussed.

Author response: Thank you for pointing out this observation. We are not sure about this, since we do observe a clear overall increase of 11.5% in 2009 compared to the level in 2008. However, of course the catch-up effect could have been even more pronounced with respect to both magnitude and duration. Thus, we included this aspect in the revised discussion (lines 220-222).

Manuscript change: In fact, an even larger and more sustained post-strike catch-up effect (11.5% in 2009) could have been expected. However, this could as well be attributed to the overall question of capacity in the healthcare system.

Minor:

8. It is unclear to me why the overall number of procedure is included in Figure 1, especially with total numbers, if the graph focuses on the discription of procedures by "specialty" per 1,000 person years and the data may be obtained from the supplemental data anyway.

Author response: We have followed the advice of the reviewer and omitted the overall outcome in Figure 1. The original figure has been moved to the supplement, since we think it visually shows how the individual sub-specialties are compared to the total.

9. The overall incidence by combining the separate incidences of males and females in Panels 2A/3A (and the subsequent panels of the Figures in S2) does not add any information, because these populations are not similar based on the sex-specific incidence and should be ommitted.

Author response: In the revised version we have omitted the panel with overall incidence. Instead, we have included a panel with the incidence in children with chronic disease.

10. The figures are not designed to be inclusively. The use of colours is certainly not colourblind-friendly and needs to be adjusted or at least supplemented by different line types.

Author response: We have followed the advice of the reviewer by adopting different line types for some of the figures and for the confidence limits.

11. Connecting individual values with a line results in the reader to assume that they are connected in any way. However, the statistical analysis assumes them to be independent, which would favour a depiction of individual data points without being connected by a line. The use of individual confidence intervals (instead of a, for example shaded region) also supports not using a connecting line.

Author response: We agree with the reviewer that the figures ought to be interpreted visually rather than analytically. Individual annual data points and regions for the confidence limits by dashed lines have been provided when visually appropriate.

12. The colour choice of the confidence intervals together with the rather low image quality makes the visual assessment of the confidence intervals quite difficult.

Author response: We hope that the revised layout enhances the visual assessment of the figures. Further, we agree that the image quality of the figures as they appear in the submission file is low, but that is not the case for the original files, and we are confident that it will neither be the case for the final manuscript if accepted for publication. Further, we have ensured that all figures comply to the PLOS requirements using the Preflight Analysis and Conversion Engine (PACE) digital diagnostic tool, https://pacev2.apexcovantage.com/.

13. Why does Figure 1 not have any confidence intervals apart from those of the overall data?

Author response: When confidence limits are omitted it is due to the visual appearance of the curves and figures. Multiple overlapping curves may only distort the assessment and interpretation. For the exact numbers, we refer to the tables in S1 file.

Reviewer #2: Very well written paper with detailed analysis.

Author response: We thank the reviewer for the complimentary assessment of our manuscript.

In line 180 and 193 - you have mentioned "decrease in children" - can you please substantiate and provide references?

Author response: We thank the reviewer for noticing this. We have provided the references to the Figures 1, 2 and 3 in the revised manuscript.

Frenulotomy - Can you please elaborate if these are performed in GA?

Author response: Unfortunately, we only have information on the procedure code and not whether local or general anaesthesia was used. This has been clarified in the methods section of the revised manuscript (lines 86-87).

Manuscript change: Information on local or general anaesthesia regarding the procedures in public hospitals was not available.

---

## [Editor Report · Decision Letter 1]

11 Apr 2023

PONE-D-22-35565R1Surgical procedures in Danish children 1999-2018PLOS ONE

Dear Dr. Jensen,

Thank you for submitting your manuscript to PLOS ONE. After careful consideration, we feel that it has merit but does not fully meet PLOS ONE’s publication criteria as it currently stands. Therefore, we invite you to submit a revised version of the manuscript that addresses the points raised during the review process.

ACADEMIC EDITOR: You implemented all amedments required by the 1st reviewer except your reply to Q4 is not fully implemented in the revised version. So, please do it.There was no conflicts between reviewsFrom my side I found the manuscript well written and interesting for readers and meet PLOS ONE'S publication criteria.==============================

We look forward to receiving your revised manuscript.

Kind regards,

Stefan Grosek, Ph.D., M.D.,

Academic Editor

PLOS ONE
---

## [Author Response · Author response to Decision Letter 1]

12 Apr 2023

Review Comments to the Author and Author response

4. It remains unclear why the study period was chosen and why procedures in private hospitals were included only with their availability in the database. Wouldn't it be the more natural choice to start the investigation in 2002 then, because lines 68-70 do not provide a strong justification for the choice of the investigated years.

Author response: The study period was chosen as 1999-2018 due to data availability and to ensure that all calendar years throughout the study period included children 0-5 years of age. Cf. line 70-73. Apart from that, however, we used the maximal study period to exploit as much information as possible.

We acknowledge that the registration of procedures in private hospitals is unclear. We have tried to specify in the revised version (lines 93-95).

Manuscript changes: The cohort was based on all Danish children liveborn in the period 1994-2018. Children were followed from birth until five years of age, 31 December 2018, migration, or death, whichever came first. The observation period was chosen as 1999 through 2018 to ensure similar age distributions across calendar years. Before 1999 only children less than 5 years of age were present in the cohort. (…) However, note that information on private hospitals was only included in The National Patient Register from 2002 onwards and due to initial lack of registrations in the subsequent years, the present study covered the period 2004-2018 for the outcome surgical procedures in private hospitals.

---

## [Editor Report · Decision Letter 2]

14 Apr 2023

Surgical procedures in Danish children 1999-2018

PONE-D-22-35565R2

Dear Dr. Jensen,

We’re pleased to inform you that your manuscript has been judged scientifically suitable for publication and will be formally accepted for publication once it meets all outstanding technical requirements.

Kind regards,

Stefan Grosek, Ph.D., M.D.,

Academic Editor

PLOS ONE
---

## [Editor Report · Acceptance letter]

19 Apr 2023

PONE-D-22-35565R2 

Surgical procedures in Danish children 1999-2018 

Dear Dr. Jensen:

I'm pleased to inform you that your manuscript has been deemed suitable for publication in PLOS ONE. Congratulations! Your manuscript is now with our production department. 

Kind regards, 

on behalf of

Professor Stefan Grosek 

Academic Editor

PLOS ONE